# Number and timing of primary cleft lip and palate repair surgeries in England: whole nation study of electronic health records before and during the COVID-19 pandemic

David Etoori,[1] Min Hae Park,[2,3] Ruth Marion Blackburn ![ORCID],[1] Kate J Fitzsimons ![ORCID],[3] Sophie Butterworth,[3] Jibby Medina,[3] Louise Mc Grath-Lone,[4] Craig Russell,[3,5] Jan van der Meulen[2,3]

For numbered affiliations see end of article.

**Correspondence to**
Dr Ruth Marion Blackburn;
r.blackburn@ucl.ac.uk

## ABSTRACT

**Objective** To quantify differences in number and timing of first primary cleft lip and palate (CLP) repair procedures during the first year of the COVID-19 pandemic (1 April 2020 to 31 March 2021; 2020/2021) compared with the preceding year (1 April 2019 to 31 March 2020; 2019/2021).

**Design** National observational study of administrative hospital data.

**Setting** National Health Service hospitals in England.

**Study population** Children <5 years undergoing primary repair for an orofacial cleft Population Consensus and Surveys Classification of Interventions and Procedures-fourth revisions (OPCS-4) codes F031, F291).

**Main exposure** Procedure date (2020/2021 vs 2019/2020).

**Main outcomes** Numbers and timing (age in months) of first primary CLP procedures.

**Results** 1716 CLP primary repair procedures were included in the analysis. In 2020/2021, 774 CLP procedures were carried out compared with 942 in 2019/2020, a reduction of 17.8% (95% CI 9.5% to 25.4%). The reduction varied over time in 2020/2021, with no surgeries at all during the first 2 months (April and May 2020). Compared with 2019/2020, first primary lip repair procedures performed in 2020/2021 were delayed by 1.6 months on average (95% CI 0.9 to 2.2 months). Delays in primary palate repairs were smaller on average but varied across the nine geographical regions.

**Conclusion** There were significant reductions in the number and delays in timing of first primary CLP repair procedures in England during the first year of the pandemic, which may affect long-term outcomes.

## STRENGTHS AND LIMITATIONS OF THIS STUDY

⇒ We analysed administrative hospital data (Hospital Episode Statistics; HES) with whole nation coverage of England for children undergoing surgical repair of cleft lip and palate (CLP) at in two time periods; before (2019/2020) or during (2020/2021) the COVID-19 pandemic.

⇒ Within these time periods, we examined the timing of first surgical repair with respect to clinical guidelines advocating surgery for first repair of cleft lip in children aged 3–6 months and by age 13 months for cleft palate repair.

⇒ To reduce the risk of misclassifying the timing of surgery, we restricted the study population to children born in hospitals in England, meaning that some children who had CLP surgery (but who did not have a birth record in HES) were excluded from the analysis.

⇒ Even though our study had whole nation coverage of England the numbers of children within some important subgroups (eg, narrower ethnic groups) were insufficient to support further analysis.

## INTRODUCTION

Around 1 in 670 children in England, Wales and Northern Ireland are born alive with an orofacial cleft that may affect only the lip, only the palate or both.[1] An orofacial cleft can have significant effects on children's lives, including ongoing hearing loss, speech and language difficulties, psychosocial difficulties and lower educational attainment.[2–7] It is recommended that children with a cleft palate have surgery to repair their cleft when they are between 6 and 12 months old as this would reduce the likelihood of negative outcomes.[8 9] Cleft lip repair procedures are usually performed when the children are between 3 and 6 months old, a time frame suggested by a handful of small studies showing that early repair leads to better aesthetic results,[10 11] improved feeding[10] and better psychosocial development.[12]

Access to healthcare declined markedly during the COVID-19 pandemic.[13 14] This decline represents both the postponement

and cancellation of planned care. For some time-sensitive procedures such as cleft lip and palate (CLP) repair, delays could have a detrimental effect on long-term outcomes.

This study aimed to quantify the impact of the COVID-19 pandemic on the number and the timing of first primary CLP repair procedures using national longitudinal administrative hospital data from the English National Health Service (NHS). We hypothesised that there would be a reduction in the number of first primary CLP repair procedures during the COVID-19 pandemic year (2020/2021) compared with the preceding year (2019/2020). We defined the start of the first COVID-19 pandemic year as 1 April 2020 as this coincides closely with the official start of the first nationwide lockdown in England on 23 March 2020.[15]

We also hypothesised that first primary CLP repair procedures would be delayed during the pandemic, so that the children at the time of surgery would be older than in the preceding year. Quantifying the extent of delays to surgery is important for planning of the future needs of these children.

## METHODS
### Study design
This is an observational study comparing the numbers of procedures, and the age at surgery for primary repair of cleft lip and/or palate at hospitals in England before (2019/2020), or during (2020/21) the COVID-19 pandemic

### Data source and study population
We used the Hospital Episode Statistics (HES), a national database including records of all episodes in NHS hospitals derived from administrative data.[16] HES records include diagnostic fields coded according to the International Classification of Diseases-10th revision (ICD-10)[17] and procedure fields coded according to the Population Consensus and Surveys Classification of Interventions and Procedures-fourth revisions (OPCS-4).[18]

We identified all children born after 1 April 2014, who were considered to have an orofacial cleft because they had both a record with relevant diagnostic codes before their second birthday (or until 31 March 2021, whatever came earlier) and a record with relevant CLP repair procedure codes before their fifth birthday (or until 31 March 2021, whatever came earlier; see online supplemental information 1 for code lists). We excluded children without a birth record in HES and children born from multiple pregnancies. Births recorded in HES represent 97% of all births in England.[19] Please see online supplemental figure 1.

### Outcome and patient characteristics
In the children identified with an orofacial cleft, we determined the date/s of their first primary CLP repair procedure, with primary lip repair and primary palate repair treated separately, such that some children

contributed more than one surgery. Secondary procedures were excluded from the analytical sample as other factors might influence their timing, including the timing of the primary surgery. We used diagnostic codes to distinguish four cleft types (see online supplemental information 2 for code lists): cleft lip only, cleft palate only, unilateral CLP, bilateral CLP. We used procedure codes to capture the type of surgery: primary lip repair (F031) and primary palate repair (F291). We also used ICD-10 codes to determine whether there were other additional congenital malformations (online supplemental information 3).[20 21] Quintiles of the national distribution of the 2019 Index of Multiple Deprivation rankings of 32 844 lower super output areas (LSOA; areas with typically 1500 inhabitants and 600 households) were used to categorise children into 5 groups according to their socioeconomic background.[22] Ethnicity was coded as white, and minority ethnicity including black, Asian, mixed race and other. Nine geographical regions of residence that correspond to the nine regionally commissioned cleft services of England were derived from the LSOA.

### Statistical analyses
We counted the number of first primary CLP repair procedures in 2020/2021 and 2019/2020 and calculated the relative difference between these numbers. CIs for these relative differences were calculated using the conditional method for testing differences between two Poisson means.[23] We used the Mantel-Haenszel test of homogeneity to investigate whether the difference between the number of procedures in 2019/2020 and 2020/2021 varied according to the children's characteristics.

To investigate changes in timing of first primary CLP repair procedures, we compared the mean age at the time of the first primary CLP procedures carried out in 2019/2020 and 2020/2021 with the t-test. Linear regression with interaction terms was used to test whether the difference between the means in 2019/2020 and 2020/2021 varied according to the children's characteristics.

Children with missing data on a specific characteristic were not included in the analyses involving that characteristic. A p<0.05 was considered to indicate a statistically significant result. All analyses were performed in Stata V.17 (Statacorp).[24]

### Patient and public involvement
The Education and Child Health Insights from Linked Data (ECHILD) project undertakes regular patient and public involvement (PPI) including the acceptability of the use of deidentified data from healthcare and education settings, and research priorities for these datasets. Children and parents in our PPI workshops identified understanding the health and education impact of the pandemic on children with additional clinical needs (such as CLP) as a key priority for research.

# RESULTS

## Study population

We identified 6438 children with a CLP procedure code recorded before the age of 5 between 1 April 2014 and 31 March 2021. Of these, 680 (10.6%) did not have a birth record or were born from multiple pregnancies and 257 (4.0%) did not have a CLP diagnostic code recorded before the age of 2. These children were, therefore, excluded (online supplemental figure 1).

## Number of first primary CLP repair procedures during the COVID-19 pandemic

In the remaining 5501 children, we identified 774 first primary CLP procedures in 2020/2021 corresponding to 321 first lip repair and 453 first palate repairs. This was in comparison to 942 procedures (408 lip repairs, 534 palate repairs) in 2019/2020, a reduction of 17.8% (95% CI 9.5% to 25.4%; p<0.001; table 1).

The reduction in the number of first lip repair observed in 2020/2021 did not vary significantly according to the children's characteristics (p always>0.1 for cleft type, presence of additional anomalies, deprivation or ethnicity) or geographical region of residence. However, the reduction in lip repairs did vary according to quarterly period (p<0.0001).

The reduction in the number of the first primary palate repair procedures in 2020/2021 varied according to quarterly period (p<0.0001) and was significantly larger for children with additional congenital malformations (p=0.0210).

No repair procedures were carried out in the first 2 months of the study period (1 April 2020 to 31 May 2020), primary cleft surgery resumed in the third month of the first quarter. The numbers of first primary procedures undertaken in the second and third quarters of 2020/2021 (1 July 2020 and 31 December 2020) were higher and the number in the fourth quarter (between 1 January and 31 March) was lower than in the corresponding months in the preceding year (figure 1).

## Timing of CLP surgeries before and during the COVID-19 pandemic

The mean age at the first primary lip repairs increased by 1.6 months (95% CI 0.9 to 2.2) in the first year of the COVID-19 pandemic compared with 2019/2020 (see also figure 2). This increase in age did not vary according to the children's characteristics (p always>0.1). The largest increases in mean age of lip repairs were in the South-West 3.9 months (95% CI 2.7 to 5.1), the East Midlands 3.5 months (95% CI 2.0 to 4.9) and the first quarter of 2020/2021 3.4 months (95% CI 0.14 to 6.7) (table 2).

At national level, mean age at the first primary palate repair did not increase during 2020/2021 (0.6 months, 95% CI −0.2 to 1.4) but there was some evidence of regional variation (p=0.0022) with the largest increases in mean age being observed in the South-West (5.6 months; 95% CI 3.2 to 7.9) and North-West (2.6 months; 95% CI 0.5 to 4.6).

There was an increase in the proportion of lip repairs carried out after the age of 6 months from 19.4% (79/408; 95% CI 15.6% to 23.5%) in 2019/2020 to 57.9% (186/321; 95% CI 52.3% to 63.4%) in 2020/2021 (p<0.0001). There was also a small but significant increase in the proportion of palate repairs carried out after the age of 12 months from 22.5% (120/534; 95% CI 19.0% to 26.2%) in 2019/2020 to 28.7% (130/453; 95% CI 24.6% to 33.1%) in 2020/2021 (p=0.025).

# DISCUSSION

This national study using routinely collected administrative hospital data of children born with an orofacial cleft in England found an 18% reduction in the number of first primary CLP repair procedures during the first year of the COVID-19 pandemic, as well as a delay of 1.6 months in the timing of the first primary lip repair procedure, compared with the preceding year. The largest difference was observed during the first quarter of the COVID-19 pandemic period. Also, the delay in the timing of procedures varied across the country with children residing in the South-West most affected.

The study has several strengths. First, the study population had excellent geographical coverage of England, reflecting all NHS hospitals. Second, by using both diagnosis and procedure codes to identify the study population, the impact of coding errors on the differences reported will have been reduced. Third, the relatively large study population made it possible to report differences in number and timing of first primary CLP repair surgeries undertaken by patient characteristics, by region and quarterly period.

Limitations include that for some children data items on their specific diagnosis were missing and when differences were compared by the children's characteristics, our results were based only on a complete-case analysis. It is unclear what impact this may have on the results reported as it is not known whether children with missing data were more or less likely to have delayed surgery for CLP repair than those with complete data. The use of ICD-10 and OPCS-4 codes may not capture more nuanced clinical information about individual diagnoses and procedures.

We showed that CLP repair surgery completely stopped in April and May 2020, which coincides with the start of the first national 'lockdown' in England on 23rd March 2020. This translated to a reduction in numbers of both primary lip repairs and primary palate repairs. Stakeholders will need to continue monitoring this as these reductions could have long-term consequences (eg, on speech development) and may have a time lag in their effect. The reduction in the number of first palate repair surgeries for children with additional congenital malformations was larger than for children without additional malformations, which may reflect deferred surgeries for children at higher risk of complications from

Table 1  Number of first primary cleft lip and palate repair procedures by year of surgery and the children's characteristics

| | No of procedures | | | | Palate repairs | | | |
|---|---|---|---|---|---|---|---|---|
| | Lip repair | | | | | | | |
| Year of surgery* | 2019/2020 | 2020/2021 | Relative difference (95% CI) | P value | 2019/2020 | 2020/2021 | Relative difference (95% CI) | P value |
| All | 408 | 321 | −21.32 (−32.24 to −8.71) | 0.0013 | 534 | 453 | −15.17 (−25.32 to −3.67) | 0.0099 |
| Cleft type | (p=0.6389)† | | | | (p=0.8277)† | | | |
| Cleft lip only | 157 | 116 | −26.11 (−42.39 to −5.47) | 0.0131 | – | – | – | – |
| Cleft palate only | – | – | – | – | 284 | 233 | −17.96 (−31.31 to 2.09) | 0.0249 |
| Unilateral CLP | 165 | 141 | −14.54 (−32.24 to 7.65) | 0.1707 | 164 | 147 | −10.37 (−28.75 to 12.67) | 0.3358 |
| Bilateral CLP | 86 | 64 | −25.58 (−47.01 to 4.05) | 0.073 | 86 | 73 | −15.12 (−38.73 to 17.32) | 0.3041 |
| Congenital malformations | (p=0.1867)† | | | | (p=0.0210)† | | | |
| No additional malformations | 282 | 207 | −26.60 (−38.95 to −11.86) | 0.0007 | 240 | 237 | −1.25 (−17.82 to 18.66) | 0.8909 |
| Additional malformations | 126 | 114 | −9.52 (−30.39 to 17.50) | 0.4396 | 294 | 216 | −26.53 (−38.65 to −12.12) | 0.0005 |
| IMD quintile | (p=0.9045)† | | | | (p=0.8962)† | | | |
| Q1 (Most deprived) | 94 | 86 | −8.51 (−32.51 to 23.90) | 0.5522 | 127 | 114 | −10.24 (−30.91 to 16.51) | 0.4034 |
| Q2 | 79 | 61 | −22.78 (−45.64 to 9.22) | 0.1293 | 117 | 98 | −16.24 (−36.62 to 10.49) | 0.196 |
| Q3 | 65 | 53 | −18.46 (−44.36 to 19.02) | 0.2712 | 92 | 78 | −15.22 (−38.11 to 15.90) | 0.2843 |
| Q4 | 61 | 45 | −26.23 (−50.95 to 10.23) | 0.1215 | 62 | 51 | −17.74 (−44.35 to 21.11) | 0.3029 |
| Q5 (least deprived) | 47 | 36 | −23.40 (−51.79 to 20.79) | 0.2299 | 55 | 57 | 3.51 (−42.28 to 34.61) | 0.8509 |
| Missing | 62 | 40 | | | 81 | 55 | | – |
| Ethnicity | (p=0.7415)† | | | | (p=0.3669)† | | | |
| White/white British | 327 | 253 | −22.63 (−34.60 to −8.55) | 0.0021 | 414 | 341 | −17.63 (−28.84 to −4.71) | 0.0079 |
| Minority ethnicity | 74 | 61 | −17.57 (−42.25 to 17.28) | 0.2649 | 112 | 106 | −5.36 (−28.12 to 24.55) | 0.6852 |
| Missing | 7 | 7 | – | – | 8 | 6 | – | – |
| Region | (p=0.4821)† | | | | (p=0.3715)† | | | |
| North-East | 25 | 20 | −8.70 (−51.94 to 72.56) | 0.766 | 20 | 30 | 34.37 (−17.37 to 64.03) | 0.1337 |
| North-West | 45 | 50 | 6.12 (−43.34 to 38.61) | 0.7596 | 75 | 60 | −18.42 (−42.63 to 15.61) | 0.2349 |
| Yorkshire | 40 | 30 | −21.05 (−52.76 to 30.85) | 0.3356 | 55 | 40 | −28.30 (−54.01 to 10.82) | 0.1174 |
| East Midlands | 35 | 30 | −14.29 (−49.17 to 43.71) | 0.5386 | 35 | 25 | −27.78 (−58.13 to 22.98) | 0.2074 |
| West Midlands | 30 | 30 | −3.12 (−42.83 to 63.95) | 0.9007 | 45 | 50 | 9.80 (−37.07 to 40.79) | 0.6137 |
| East of England | 30 | 25 | −28.12 (−59.84 to 26.73) | 0.2288 | 40 | 30 | −28.57 (−56.83 to 16.87) | 0.1597 |
| London | 50 | 35 | −28.57 (−55.07 to 12.49) | 0.1284 | 60 | 60 | 0 (−45.48 to 31.26) | 1.0000 |
| South-East | 50 | 45 | −9.61 (−40.40 to 36.74) | 0.6173 | 65 | 65 | −1.54 (−31.38 to 41.24) | 0.9302 |
| South-West | 35 | 15 | −57.14 (−78.25 to −19.47) | 0.0046 | 40 | 30 | −21.95 (−52.42 to 27.01) | 0.2954 |

Continued

**Table 1** Continued

| Year of surgery* | No of procedures | | | | | | | |
|---|---|---|---|---|---|---|---|---|
| | Lip repair | | | | Palate repairs | | | |
| | 2019/2020 | 2020/2021 | Relative difference (95% CI) | P value | 2019/2020 | 2020/2021 | Relative difference (95% CI) | P value |
| Missing | 65 | 40 | – | – | 95 | 60 | – | – |
| Quarter | (p<0.0001)† | | | | (p<0.0001)† | | | |
| Q1 (April–June) | 105 | 10 | –90.48 (–95.56 to –81.78) | <0.0001 | 136 | 27 | –80.15 (–87.38 to –69.83) | <0.0001 |
| Q2 (July–September) | 107 | 110 | 2.73 (–26.16 to 28.10) | 0.839 | 149 | 182 | 18.13 (–2.22 to 34.52) | 0.07 |
| Q3 (October–December) | 103 | 140 | 26.43 (4.43 to 43.52) | 0.0177 | 123 | 130 | 5.38 (–22.01 to 26.66) | 0.6606 |
| Q4 (January–March) | 93 | 61 | –34.41 (–53.31 to –8.43) | 0.0099 | 126 | 114 | –9.52 (–30.39 to 17.50) | 0.4396 |

Region figures rounded to the nearest 5 for disclosure control.
*2020/2021: first year of COVID-19 pandemic; 2019/2020: preceding year.
†Mantel-Haenszel test for homogeneity, testing if the relative differences vary according to the children's characteristics.
CLP, cleft lip and palate; IMD, Index of Multiple Deprivation .

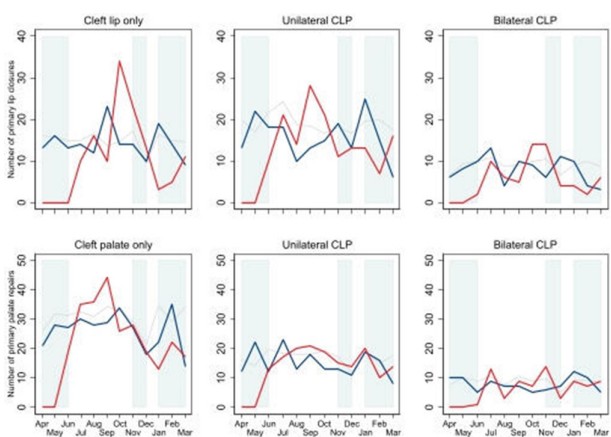

**Figure 1** Monthly numbers of first primary cleft lip repair and palate repair procedures in the first year of the COVID-19 period (between April 2020 and March 2021; red line) and the preceding year (between April 2019 and March 2020; blue line).Grey lines represent 5-year average (2014/2015–2018/2019) for historic comparison. Shaded areas represent lockdown periods (lockdown 1: 23 March 2020–23 June 2020; lockdown 2: 5 November 2020–6 December 2020; lockdown 3: 1 January 2021–8 March 2021). CLP, cleft lip and palate.

COVID-19.[25] We also showed an increase in the age at first lip repair surgery, but no significant increase in the age at first palate repair. This may reflect clinical prioritisation of primary cleft palate repairs over cleft lip repairs. UK national guidance suggests that palate repair should be complete by 13 months of age (3–6 months of age for lip repairs).[26 27] However, we also showed that a significantly larger proportion of children had their first palate repair surgery after 12 months, which might have long-term consequences for education attainment as children who receive palate repairs after 13 months have been shown to have less favourable speech outcomes.[8 9]

We note that birth rates for England have slightly decreased over the study period, with a proportionate

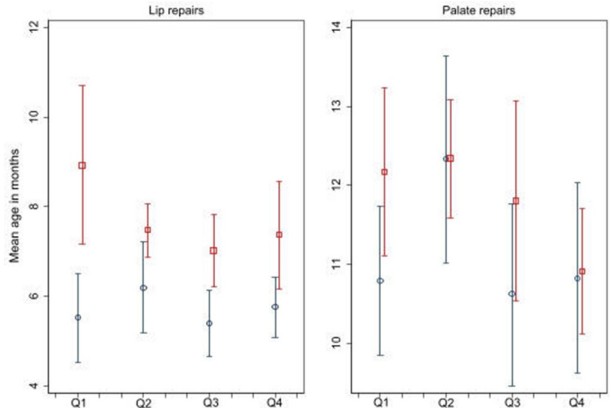

**Figure 2** Mean age at the first primary cleft lip and palate repair procedure in the first year of the COVID-19 pandemic (between April 2020 and March 2021; red square) and the preceding year (between April 2019 and March 2020; blue circle) with 95% CIs. Q1—April–June; Q2—July–September; Q3—October–December; Q4—January–March.

**Table 2** Mean age at first primary CLP repair surgery by year of surgery and exposure variables

| Year of surgery* | Mean age at surgery in months (95% CI) | | | | | | | |
| | Lip repair | | | | Palate repairs | | | |
| | 2019/2020 | 2020/2021 | Difference | P value | 2019/2020 | 2020/2021 | Difference | P value |
|---|---|---|---|---|---|---|---|---|
| All | 5.72 (5.28 to 6.15) | 7.30 (6.84 to 7.76) | 1.58 (0.94 to 2.22) | <0.0001 | 11.19 (10.60 to 11.77) | 11.81 (11.30 to 12.33) | 0.63 (−0.16 to 1.42) | 0.1209 |
| Cleft type | (p=0.4166)† | | | | (p=0.9728)† | | | |
| Cleft lip only | 5.42 (4.77 to 6.08) | 7.14 (6.22 to 8.05) | 1.71 (0.62 to 2.80) | 0.0022 | – | – | – | – |
| Cleft palate only | – | – | – | – | 12.74 (11.92 to 13.57) | 13.50 (12.74 to 14.25) | 0.75 (−0.39 to 1.88) | 0.1961 |
| Unilateral CLP | 5.46 (4.83 to 6.10) | 7.34 (6.81 to 7.87) | 1.88 (1.04 to 2.72) | <0.0001 | 8.96 (8.10 to 9.82) | 9.66 (9.08 to 10.25) | 0.70 (−0.35 to 1.76) | 0.1919 |
| Bilateral CLP | 6.75 (5.58 to 7.93) | 7.51 (6.31 to 8.70) | 0.75 (−0.94 to 2.45) | 0.3816 | 10.28 (8.73 to 11.83) | 10.77 (9.29 to 12.24) | 0.49 (−1.66 to 2.64) | 0.6525 |
| Congenital malformations | (p=0.8086)† | | | | (p=0.4014)† | | | |
| No additional malformations | 5.02 (4.68 to 5.37) | 6.56 (6.14 to 6.98) | 1.53 (0.99 to 2.07) | <0.0001 | 9.27 (8.64 to 9.90) | 10.47 (9.95 to 10.98) | 1.20 (0.38 to 2.01) | 0.0039 |
| Additional malformations | 7.27 (6.13 to 8.42) | 8.65 (7.61 to 9.68) | 1.37 (−0.17 to 2.91) | 0.0813 | 12.75 (11.86 to 13.65) | 13.29 (12.41 to 14.18) | 0.54 (−0.75 to 1.83) | 0.4102 |
| IMD quintile | (p=0.9259)† | | | | (p=0.9099)† | | | |
| Q1 (Most deprived) | 5.42 (4.58 to 6.26) | 7.29 (6.35 to 8.23) | 1.87 (0.62 to 3.12) | 0.0035 | 10.38 (9.29 to 11.48) | 11.64 (10.40 to 12.88) | 1.26 (−0.38 to 2.90) | 0.132 |
| Q2 | 6.34 (5.09 to 7.59) | 8.25 (6.68 to 9.82) | 1.91 (−0.05 to 3.87) | 0.0561 | 10.87 (9.83 to 11.91) | 11.47 (10.54 to 12.40) | 0.60 (−0.82 to 2.01) | 0.4067 |
| Q3 | 5.85 (5.02 to 6.69) | 7.18 (6.55 to 7.81) | 1.33 (0.25 to 2.40) | 0.0164 | 10.42 (9.29 to 11.55) | 11.51 (10.52 to 12.50) | 1.09 (−0.42 to 2.61) | 0.1557 |
| Q4 | 5.96 (4.52 to 7.41) | 7.06 (5.77 to 8.35) | 1.10 (−0.89 to 3.09) | 0.2772 | 11.22 (9.63 to 12.80) | 12.45 (10.62 to 14.28) | 1.23 (−1.15 to 3.62) | 0.3086 |
| Q5 (least deprived) | 4.91 (4.40 to 5.41) | 6.83 (5.95 to 7.72) | 1.93 (0.98 to 2.87) | 0.0001 | 11.78 (10.21 to 13.35) | 13.68 (12.05 to 15.32) | 1.91 (−0.34 to 4.15) | 0.0949 |
| Ethnicity | (p=0.9406)† | | | | (p=0.9348)† | | | |
| White/White British | 5.54 (5.09 to 6.00) | 7.23 (6.72 to 7.75) | 1.69 (1.00 to 2.38) | <0.0001 | 11.34 (10.64 to 12.04) | 11.94 (11.34 to 12.54) | 0.60 (−0.34 to 1.54) | 0.2129 |
| Minority ethnicity | 6.00 (5.06 to 6.93) | 7.63 (6.44 to 8.82) | 1.63 (0.15 to 3.11) | 0.0312 | 10.81 (9.80 to 11.82) | 11.49 (10.40 to 12.57) | 0.68 (−0.80 to 2.15) | 0.3659 |
| Region | (p=0.2113)† | | | | (p=0.0022)† | | | |

Continued

**Table 2** Continued

| Year of surgery* | Mean age at surgery in months (95% CI) | | | | | | | |
|---|---|---|---|---|---|---|---|---|
| | Lip repair | | | | Palate repairs | | | |
| | 2019/2020 | 2020/2021 | Difference | P value | 2019/2020 | 2020/2021 | Difference | P value |
| North-East | 4.92 (3.44 to 6.41) | 5.51 (4.30 to 6.72) | 0.59 (−1.29 to 2.47) | 0.5297 | 10.27 (8.15 to 12.39) | 9.76 (8.39 to 11.13) | −0.51 (−2.86 to 1.83) | 0.6622 |
| North-West | 5.10 (4.08 to 6.12) | 7.13 (5.80 to 8.46) | 2.03 (0.36 to 3.70) | 0.0176 | 8.45 (7.36 to 9.54) | 11.02 (9.15 to 12.88) | 2.56 (0.51 to 4.62) | 0.0146 |
| Yorkshire | 4.87 (4.08 to 5.66) | 7.00 (5.55 to 8.45) | 2.13 (0.59 to 3.66) | 0.0072 | 9.68 (8.37 to 10.99) | 11.21 (8.32 to 14.10) | 1.53 (−1.31 to 4.37) | 0.2881 |
| East Midlands | 4.71 (3.95 to 5.46) | 8.16 (6.81 to 9.51) | 3.45 (2.00 to 4.91) | <0.0001 | 9.15 (7.56 to 10.74) | 11.36 (9.30 to 13.42) | 2.21 (−0.29 to 4.72) | 0.0827 |
| West Midlands | 6.41 (4.02 to 8.80) | 8.33 (7.59 to 9.06) | 1.91 (−0.57 to 4.40) | 0.1283 | 10.34 (8.60 to 12.08) | 11.20 (10.43 to 11.96) | 0.86 (−0.95 to 2.67) | 0.3492 |
| East of England | 5.55 (4.47 to 6.64) | 5.88 (5.11 to 6.65) | 0.33 (−1.08 to 1.73) | 0.6439 | 12.12 (9.60 to 14.64) | 12.14 (10.05 to 14.22) | 0.01 (−3.40 to 3.43) | 0.9935 |
| London | 6.36 (4.50 to 8.22) | 6.88 (4.56 to 9.21) | 0.52 (−2.38 to 3.43) | 0.7203 | 12.04 (10.09 to 13.99) | 10.83 (10.02 to 11.64) | −1.21 (−3.30 to 0.87) | 0.252 |
| South-East | 6.51 (5.02 to 8.00) | 8.34 (6.84 to 9.84) | 1.83 (−0.26 to 3.92) | 0.0858 | 14.16 (12.34 to 15.97) | 13.91 (12.61 to 15.21) | −0.24 (−2.46 to 1.97) | 0.829 |
| South-West | 5.83 (5.18 to 6.48) | 9.72 (8.50 to 10.93) | 3.89 (2.66 to 5.12) | <0.0001 | 10.96 (9.59 to 12.33) | 16.51 (14.42 to 18.60) | 5.55 (3.18 to 7.92) | <0.0001 |
| Quarter | (p=0.6010)† | | | | (p=0.5800)† | | | |
| Q1 (Apr–Jun) | 5.52 (4.53 to 6.51) | 8.92 (7.15 to 10.69) | 3.40 (0.14 to 6.66) | 0.0412 | 10.79 (9.85 to 11.74) | 12.17 (11.11 to 13.24) | 1.38 (−0.80 to 3.56) | 0.2142 |
| Q2 (Jul–Sep) | 6.19 (5.18 to 7.21) | 7.47 (6.87 to 8.07) | 1.27 (0.11 to 2.44) | 0.0322 | 12.33 (11.01 to 13.64) | 12.34 (11.58 to 13.09) | 0.01 (−1.44 to 1.45) | 0.9915 |
| Q3 (Oct–Dec) | 5.39 (4.66 to 6.12) | 7.02 (6.22 to 7.82) | 1.63 (0.51 to 2.75) | 0.0046 | 10.62 (9.46 to 11.77) | 11.80 (10.54 to 13.07) | 1.19 (−0.52 to 2.90) | 0.1721 |
| Q4 (Jan–Mar) | 5.76 (5.08 to 6.43) | 7.37 (6.16 to 8.57) | 1.61 (0.34 to 2.88) | 0.0133 | 10.82 (9.62 to 12.03) | 10.91 (10.11 to 11.71) | 0.08 (−1.38 to 1.55) | 0.9093 |

*2020/2021: first year of COVID-19 pandemic; 2019/2020: preceding year.
†Test for interaction testing to see if the differences in mean age vary according to the children's characteristic.
CLP, cleft lip and palate; IMD, Index of Multiple Deprivation.

decline in the number of children born with CLP.[28] This may have had a small impact on the number of expected operations but does not fully explain the observed reduction in number of procedures. The number of registrations recorded in the Cleft Registry and Audit Network (CRANE) database for children born with cleft was approximately 7.7% lower in 2021 compared with 2020,[29] which is not sufficient to explain the relative reduction observed in our study (17.8% reduction). Furthermore, we observed delays in the timing of the lip and palate repairs (which is not dependent on the number of children born with a cleft), although the difference observed for palate repairs was not statistically significant.

Our study indicated that there were regional variations in the impact of the COVID-19 pandemic on the timing of first primary CLP repair procedures, which may reflect differences in the regions' influence on management decision-making, resources, fragility and capacity for recovery. The delay of almost 6 months seen in one region with other regions showing hardly any delay in the timing of first primary CLP repair procedures requires further investigation. COVID-19 pandemic-associated delays in CLP repair have been reported in other countries, including a single-centre study in Peru where 172 patients demonstrated increases in age at the time of primary lip and palate repair.[30] Similarly, reduced volumes of procedures were recorded during the pandemic (relative to the prepandemic period) for low-income and middle-income countries reporting to the Smile Train Express platform.[31]

This paper follows on from previous work which showed reductions in planned care during the pandemic and acts as a deeper dive into one specific type of planned care.[32] This work focuses on primary procedures which were given some prioritisation during the pandemic and as such might downplay its effect on wider cleft services. For example, secondary procedures such as alveolar bone graft and secondary speech surgery are time sensitive but have less evidence supporting them. While the Federation of Specialist Surgical Associations Clinical Guide to Surgical Prioritisation during the COVID-19 pandemic gave similar priority to primary and secondary cleft procedures, shop floor practicality may not necessarily have allowed equal treatment.[33] Further work needs to be done to understand the full effect of the pandemic on all cleft surgery especially the more temporally sensitive secondary cleft procedures (alveolar bone grafting and secondary speech surgery).

In conclusion, during the first year of the COVID-19 pandemic, a larger proportion of children had their cleft repair surgery outside of the recommended time frame (3–6 months for lip repair and by 13 months for palate repair). Previous research has shown that late surgery may be associated with delays in speech development and the need for additional speech therapy.[8 9] Delayed surgery beyond 13 months is thought to affect articulation following cleft palate repair, and the resulting need for extra corrective speech therapy may contribute to additional absence from school, potentially affecting primary educational attainment.[7 34] Future research should, therefore, consider investigating the effect of delay in surgery on educational outcomes to model the long-term implications of the COVID-19 pandemic.

**Author affiliations**
[1]Institute of Health Informatics, University College London, London, UK
[2]Department of Health Services Research and Policy, London School of Hygiene & Tropical Medicine, London, UK
[3]Clinical Effectiveness Unit, Royal College of Surgeons of England, London, UK
[4]University College London Institute of Education, London, UK
[5]National Cleft Surgical Service for Scotland, Royal Hospital for Children, Glasgow, UK

**Acknowledgements** We thank Professor Ruth Gilbert and Professor Katie Harron for their support of the ECHILD project.

**Contributors** JvdM and MHP developed the research question. DE, LMG-L and RMB operationalised the research question jointly with JvdM and MHP using data available via UCL Child Health Informatics team. DE undertook the analysis and drafted the manuscript. KJF, SB, JM and CR provided essential clinical and contextual input into the study design and analysis, and interpretation of the results. All authors provided oversight and input to the final manuscript. JvdM is guarantor for the study.

**Funding** This project was funded by the National Institute for Health and Care Research (NIHR) Policy Research Programme. This research was supported in part by the NIHR Great Ormond Street Hospital Biomedical Research Centre and the Health Data Research UK (grant no: LOND1), which is funded by the UK Medical Research Council and eight other funders. RMB is supported by a UKRI Innovation Fellowship funded by the Medical Research Council (grant number MR/S003797/1).

**Disclaimer** The views expressed are those of the author(s) and not necessarily those of the NIHR or the Department of Health and Social Care.

**Competing interests** None declared.

**Patient and public involvement** Patients and/or the public were not involved in the design, or conduct, or reporting, or dissemination plans of this research.

**Patient consent for publication** Not applicable.

**Provenance and peer review** Not commissioned; externally peer reviewed.

**Data availability statement** Data may be obtained from a third party and are not publicly available. Data are available on request from NHS England (formerly NHS Digital) and may not be shared by the authors.

**ORCID iDs**
Ruth Marion Blackburn http://orcid.org/0000-0002-3491-7381
Kate J Fitzsimons http://orcid.org/0000-0003-2385-3172

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
