## [Reviewer comments · BMJ Open]

This paper was submitted to a another journal from BMJ but declined for publication following peer review. The authors addressed the reviewers' comments and submitted the revised paper to BMJ Open. The paper was subsequently accepted for publication at BMJ Open.

ARTICLE DETAILS

TITLE (PROVISIONAL)	Number and timing of primary cleft lip and palate repair surgeries in England: whole nation study of electronic health records before and during the COVID-19 pandemic.
AUTHORS	Etoori, David; Park, Min Hae; Blackburn, Ruth; Fitzsimons, Kate; Butterworth, Sophie; Medina, Jibby; Mc Grath-Lone, Louise; Russell, Craig; van der Meulen, Jan

VERSION 1 – REVIEW

REVIEWER	Rossell-Perry, Percy
REVIEW RETURNED	11-Feb-2023

GENERAL COMMENTS	Dear authors, I like to suggest some comments regarding your interesting study. Method section. Type of study should be better specified. Discussion. How the number of children born with cleft may affect the estimation of the operation rates between these two periods? Is there any correlation between the delay in operating time and the clinical outcome? Are the outcomes relevant in terms of aesthetic/functional quality of the outcomes? The study may be significant improved by the inclusion of associated complication report during this pandemic. Suggested reference: Rossell-Perry P, Gavino-Gutierrez A. Cleft Lip and Palate Surgery during COVID-19 Pandemic. Plast Reconstr Surg Glob Open. 2021 Jun 29;9(6):e3692. doi: 10.1097/GOX.0000000000003692. PMID: 34235042; PMCID: PMC8245116.
--

REVIEWER	Donkor, Peter Kwame Nkrumah University of Science and Technology, College of Health Sciences, Kumasi, Ghana
REVIEW RETURNED	11-Mar-2023

GENERAL COMMENTS	This is a well written paper that documents a significant reduction in the number of primary cleft repair procedures in England during the peak of the COVID-19 pandemic. The related increase in the age of repair of the clefts have implications for long-term outcomes in the rehabilitation of affected patients, who thus require close follow up.
---

VERSION 1 – AUTHOR RESPONSE

Reviewer: 1

Method section.

Type of study should be better specified.

Author response: we have clarified the study type as follows.

“This is an observational study comparing the numbers of procedures, and the age at surgery for primary repair of cleft lip and/or palate at hospitals in England before (2019/20), or during (2020/21) the COVID-19 pandemic.”

Discussion.

How the number of children born with cleft may affect the estimation of the operation rates between these two periods?

Author response: thank you for raising this query. We have now added a section to the discussion reflecting the likely impact of changing birth rates (including those born with cleft lip and palate).

“We note that birth rates for England have slightly decreased over the study period, with a proportionate decline in the number of children born with CLP (ONS, 2022). This may have had a small impact on the number of expected operations but does not fully explain the observed reduction in number of procedures. The number of registrations recorded in CRANE for children born with cleft was approximately 7.7% lower in 2021 compared to 2020 (CRANE Database 2022 Annual Report), which is not sufficient to explain the relative reduction observed in our study (17.8% reduction). Furthermore, we observed delays in the timing of the lip and palate repairs (which is not dependent on the number of children born with a cleft), albeit the difference observed for palate repairs was not statistically significant.”

References:

Office for National Statistics, Births in England and Wales: 2021, August 2022.

<https://www.ons.gov.uk/peoplepopulationandcommunity/birthsdeathsandmarriages/livebirths/bulletins/birthsummarytablesenglandandwales/2021> (accessed 17 April 2022)

CRANE Database 2022 Annual Report, UK NHS Cleft Development Group, CRANE project team at the Clinical Effectiveness Unit, of the Royal College of Surgeons of England. https://www.crane-database.org.uk/content/uploads/2022/12/CRANE-2022-Annual-Report_12Dec22.pdf

Is there any correlation between the delay in operating time and the clinical outcome? Are the outcomes relevant in terms of aesthetic/functional quality of the outcomes?

Author response: this is a highly interesting question, that is unfortunately beyond the scope of the current paper. However, we have included some references in paragraph 1 of the introduction for studies which have shown a relationship with speech and language outcomes, aesthetic results, feeding and psychosocial outcomes (References 8-10).

As no immediate post-operative complications are recorded within the national cleft registry and audit network data (CRANE) it was not possible to explore differences in short-term outcomes. However, we are in the process of analysing linked health and education data to quantify education outcomes for children at age 5 as part of a separate research paper that uses different data and methods.

The study may be significantly improved by the inclusion of associated complication report during this pandemic. Suggested reference:

Rossell-Perry P, Gavino-Gutierrez A. Cleft Lip and Palate Surgery during COVID-19 Pandemic. *Plast Reconstr Surg Glob Open*. 2021 Jun 29;9(6):e3692. doi: 10.1097/GOX.0000000000003692. PMID: 34235042; PMCID: PMC8245116.

Author response: thank you for suggesting the inclusion of this paper, which has now been added to the discussion.

“COVID-19 pandemic associated delays in CLP repair have been reported in other countries, including a single-centre study in Peru where 172 patients demonstrated increases in age at the time of primary lip and palate repair (Rossell-Perry et al. 2021). Similarly, reduced volumes of procedures were recorded during the pandemic (relative to the pre-pandemic period) for Low and Middle Income Countries reporting to the Smile Train Express platform (Vander Burg et al. 2021)”

References:

Rossell-Perry P, Gavino-Gutierrez A. Cleft Lip and Palate Surgery during COVID-19 Pandemic. *Plast Reconstr Surg Glob Open*. 2021 Jun 29;9(6):e3692. doi: 10.1097/GOX.0000000000003692.

Vander Burg R, Agrawal K, Desai P, Desalu I, Donkor P. Impact of COVID-19 on elective cleft surgery in low- and middle-income countries. *Plastic and Reconstructive Surgery – Global Open*. 2021; 9(6): pe3656. DOI: 10.1097/GOX.0000000000003656

Reviewer: 2

Comments to the Author:

This is a well written paper that documents a significant reduction in the number of primary cleft repair procedures in England during the peak of the COVID-19 pandemic. The related increase in the age of repair of the clefts have implications for long-term outcomes in the rehabilitation of affected patients, who thus require close follow up.

Author response: thank you for your review and comments.